# Incidence and Predictors of Permanent Pacemaker Implantation after Transcatheter Aortic Valve Procedures: Data of The Netherlands Heart Registration (NHR)

**DOI:** 10.3390/jcm11030560

**Published:** 2022-01-23

**Authors:** Justine M. Ravaux, Sander M. J. Van Kuijk, Michele Di Mauro, Kevin Vernooy, Elham Bidar, Arnoud W. Van’t Hof, Leo Veenstra, Suzanne Kats, Saskia Houterman, Jos G. Maessen, Roberto Lorusso

**Affiliations:** 1Department of Cardio-Thoracic Surgery, Heart and Vascular Centre, Maastricht University Medical Centre (MUMC), 6202 AZ Maastricht, The Netherlands; sander.van.kuijk@mumc.nl (S.M.J.V.K.); mdimauro1973@gmail.com (M.D.M.); elham.bidar@mumc.nl (E.B.); Suzanne.kats@mumc.nl (S.K.); j.g.maessen@mumc.nl (J.G.M.); roberto.lorussobs@gmail.com (R.L.); 2Department of Cardiology, Maastricht University Medical Centre (MUMC), 6202 AZ Maastricht, The Netherlands; kevin.vernooy@mumc.nl (K.V.); arnoud.vant.hof@mumc.nl (A.W.V.H.); l.veenstra@mumc.nl (L.V.); 3Cardiovascular Research Institute Maastricht (CARIM), Maastricht University Medical Centre, 6202 AZ Maastricht, The Netherlands; 4Department of Cardiology, Radboud University Medical Centre (Radboudumc), 6525 GA Nijmegen, The Netherlands; 5Department of Cardiology, Zuyderland Medical Center, 6419 PC Heerlen, The Netherlands; 6Netherlands Heart Registration (NHR), 1105 AZ Amsterdam, The Netherlands; saskia.houterman@nederlandsehartregistratie.nl

**Keywords:** permanent pacemaker implantation, aortic stenosis, transcatheter aortic valve implantation

## Abstract

Atrioventricular conduction disturbance leading to permanent pacemaker (PM) implantation is a frequent and relevant complication after transcatheter aortic valve implantation (TAVI). We aimed to evaluate the rate of post-TAVI permanent PM implantation over time and to identify the predictive factors for post-TAVI PM. The data were retrospectively collected by the Netherlands Heart Registration (NHR). In total, 7489 isolated TAVI patients between 2013 and 2019 were included in the final analysis. The primary endpoint was a permanent PM implantation within 30 days following TAVI. The incidence of the primary endpoint was 12%. Post-TAVI PM showed a stable rate over time. Using multivariable logistic regression analysis, age (OR 1.01, 95% CI 1.00–1.02), weight (OR 1.00, 95% CI 1.00–1.01), creatinine serum level (OR 1.15, 95% CI 1.01–1.31), transfemoral TAVI approach (OR 1.34, 95% CI 1.11–1.61), and TAVI post-dilatation (OR 1.58, 95% CI 1.33–1.89) were shown to be independent predictors of PM. Male sex (OR 0.80, 95% CI 0.68–0.93) and previous aortic valve surgery (OR 0.42, 95% CI 0.26–0.69) had a protective effect on post-TAVI PM. From a large national TAVI registry, some clinical and procedural factors have been identified as promoting or preventing post-TAVI PM. Further efforts are required to identify high-risk patients for post-TAVI PM and to reduce the incidence of this important issue.

## 1. Introduction

Transcatheter aortic valve implantation (TAVI) is currently the first-line therapy for patients with severe aortic stenosis who are at intermediate to high surgical risk for an unfavorable post-procedural outcome [1]. Despite an ongoing trend to expand TAVI to younger and lower-risk patients, occurrence of peri-procedural conduction disturbances that lead to permanent pacemaker (PM) implantation remains a relevant shortcoming yet to be solved or substantially reduced [2,3]. A recent meta-analysis showed an average rate of PM at discharge after TAVI of 12.5% (ranging from 6.2% to 32.8%) [2]. Due to the proximity of the atrioventricular conduction system to the aortic valve structures, any intervention, either transcatheter or surgical at the valve level, may result in atrioventricular (AV) conduction system disturbances that can lead to permanent PM implantation [3]. The inherent features of the TAVI-related valve structure and deployment, however, make such an adverse event more frequent after transcatheter than surgical procedures, where post-operative PM rate reaches from 2 to 6% [4]. To some extent, the predictive factors of post-TAVI PM have been studied in the TAVI population, including pre-procedural right bundle branch block or left bundle branch block, use of self-expanding bioprosthesis, and valve implantation depth [3,5]. Larger aortic annulus, male sex, and intra-procedural AV-block have also been found to be independent predictive factors of peri-operative PM [3,6]. Despite the rather frequent incidence of post-TAVI PM, the available data in this setting are limited and inconsistent, mainly derived from relatively small population studies, registry data, and non-randomized trials [1,7]. Furthermore, post-TAVI PM is relevant as it is associated with a higher mortality at 1 year, an increased length of hospital-stay, rehospitalizations, and related cost burdens [2]. Progress in reducing the incidence of procedure-related atrioventricular conduction abnormalities leading to PM appear crucial. In this study, we aimed to assess the rate of post-TAVI PM over time as registered in a large national database, and sought to identify the patient- and procedure-related predictive factors for post-TAVI PM.

## 2. Materials and Methods

The NHR is a nationwide registry that registers fundamental pre-, operative- and post-procedural- (including follow-up) data related to the cardiac interventions performed in 16 Dutch heart centers. The data collection and registration are performed by the participating centers in a secured online environment. The aim of the NHR is to evaluate current practices in the treatment of heart disease, through all stages of the treatment process: from diagnosis to many years after the intervention. For this study, information related to the patients undergoing TAVI was collected. This study complies with the Declaration of Helsinki and the use of the data for these purposes was approved by the Maastricht University Medical Centre Ethical Committee (METC 2020-1528).

In the current study, we included all of the adult patients treated with an isolated TAVI between 1 January 2013 and 1 January 2019. The exclusion criteria included the presence of cardiac congenital pathologies, pre-operative PM, and concomitant procedures. The primary endpoint was a permanent PM implantation within the first 30 days post-TAVI. 

The continuous variables were described as mean ± standard deviation (SD) or median (range), depending on their normality. The categorical variables were presented as frequencies with percentages. The variables were largely complete with <5% missing per variable. 

To allow for the inclusion of all patients for the regression analysis, we used stochastic regression imputation with fully conditional specification to impute the dataset. The imputations were drawn using predictive mean matching.

First, a univariable logistic regression analysis was performed, with PM status as a dependent variable. The following variables were considered potential predictors of PM: sex, age, weight, creatinine serum level, diabetes mellitus, left ventricular ejection fraction, systolic pulmonary pressure, a history of lung disease, peripheral vascular disease, previous cardiac surgery, recent myocardial infarction, dialysis, Euroscore II, previous aortic valve surgery, TAVI access, pre-dilatation and post-dilatation. Subsequently, a multivariable logistic regression was carried out; once with all of the potential predictor variables (fully adjusted model), and once with a backward stepwise elimination on all of the potential predictors, to arrive at a model with only significant independent predictors. In addition, the trend of PM over time was also analyzed using univariable regression analysis. A *p*-value of <0.05 was considered statistically significant. All analyses were performed using SPSS v26 (IBM Corpn Armonk, New York, NY, USA).

## 3. Results

During the study period, 19,546 adult patients (age ≥ 18 years) were hospitalized in the Netherlands with a diagnosis of aortic stenosis requiring surgical or transcatheter aortic valve replacement. From these, 9646 underwent an isolated TAVI procedure. From the total TAVI group, 2157 patients were excluded from the final analysis due to missing data regarding their PM status, leaving 7489 patients for analysis. The flowchart of the selection of the study population is described in Figure 1.

The baseline characteristics of the patients are described in Table 1 and the procedural characteristics are recorded in Table 2. A transfemoral approach was the most frequent approach, performed in 78.7% of the patients, followed by transapical (7.7%), direct transaortic (7.6%), and trans-subclavian (5.7%) accesses. The incidence of the primary endpoint was 12%. Between 2013 and 2019, the rate of the primary endpoint varied over time, with a post-TAVI PM rate of 12% in 2013 that decreased to 11% in 2019 with a peak in 2014 and 2015 at 14% (Figure 2). After logistic regression, this trend was not significant (OR 0.960, 95% CI 0.92 to 1.00, *p* = 0.066).

By univariable analysis (Appendix A), the following patient-related variables were significantly associated with a higher risk of post-TAVI PM: weight (OR 1.01, 95% CI 1.00 to 1.01) and creatinine serum level (OR 1.15, 95% CI 1.05 to 1.25). The procedure-related factors associated with a higher risk of post-TAVI PM were transfemoral approach (OR 1.35, 95% CI 1.13–1.61) and TAVI post-dilatation (OR 1.62, 95% CI 1.37 to 1.93). Previous aortic valve surgery (OR 0.42, 95% CI 0.26 to 0.67) and male gender (OR 0.77, 95% CI 0.67 to 0.89) had a protective effect with respect to post-TAVI PM (OR 0.42, 95% CI 0.26 to 0.67).

Using multivariable or adjusted analysis (Table 3), the creatinine serum level was the only patient-related factor identified to be an independent predictor for post-TAVI PM (OR 1.15, 95% CI 1.01 to 1.31). Regarding procedure-related factors, a transfemoral approach (OR 1.34, 95% CI 1.11 to 1.61) and valve post-dilatation (OR 1.58, 95% CI 1.33 to 1.89) were confirmed as independent predictors for post-operative PM. Previous aortic valve surgery (OR 0.42, 95% CI 0.26 to 0.69) and male gender (OR 0.80, 95% CI 0.68 to 0.93) had a protective effect with respect to post-TAVI PM

By using the backward stepwise regression (Appendix A), five determinants were identified as leading to a higher rate of post-TAVI PM: age and weight of the patients, pre-procedural higher creatinine serum levels, a transfemoral TAVI approach, and TAVI aortic valve post-dilatation. Previous aortic valve surgery and male gender were found to have a protective effect on post-TAVI PM.

## 4. Discussion

The findings of the current study can be summarized as follows: (i) between 2013 and 2019, 12% of patients undergoing an isolated TAVI procedure in the Netherlands underwent post-TAVI PM; (ii) five potential independent predictive factors for PM after TAVI were identified: age and weight of the patients, a pre-procedural higher creatinine serum level, a transfemoral TAVI approach, and TAVI aortic valve post-dilatation; (iii) male gender and previous aortic valve surgery had a protective effect on post-TAVI PM; (iv) the incidence of post-TAVI PM in the Netherlands remained stable over time during the study period. 

In our study, which represents one of the largest national TAVI registries in Europe, the crude incidence of PM after TAVI was in accordance with the Belgium and Spanish registries [8,9]. In contrast, Deharo and colleagues [10] reported a higher post-operative 30-day PM rate of 20.5% after receiving a balloon-expandable valve, and 25.9% after receiving a self-expandable device, as noted in the French registry. The data from randomized control trials in selected high-risk and intermediate-risk patients showed a lower rate of post-TAVI PM [11,12].

The predictors of PM after a TAVI procedure have been recently investigated and reported [13,14]. The recent guidelines on cardiac pacing and resynchronization therapy [14] identified an association between male gender and a higher risk of post-operative PM. However, Luke and colleagues [13], in their population of 140 patients receiving a self-expandable prosthesis, showed a higher rate of PM in those of male gender, but without statistical significance. Indeed, as male patients have a larger aortic annulus [14], we can speculate that oversizing is applied less frequently in men, which may have a positive impact on atrioventricular conduction [6].

Being aged >75 years-old was already identified as a predictive factor for post-TAVI PM [15]. The need for post-TAVI PM in older patients can probably be related to additional comorbidities [16]. Indeed, advanced myocardial fibrosis or more severe calcifications of the treated aortic valve [17] may predispose a patient to a cardiac conduction system injury due to the intervention.

In TAVI patients, an “obesity paradox”, with higher BMI apparently associated with improved outcomes and survival, has been described [18]. Such an apparent “protective” factor was not confirmed in the present study. Indeed, obesity has been identified as a predictive factor for post-operative PM in several investigations [19,20]. Furthermore, the recent analysis from the STS/ACC TVT registry [20] has demonstrated that conduction disturbances leading to post-operative PM were more common in class III obese patients. The underlying mechanism of overweight promoting post-operative PM is still unclear [18,19]. A possible, more difficult procedure in obese patients undergoing a TAVI, may be due to a more approximative implant; thereby, potentially predisposing the patient to a higher risk of conduction system injury [19].

Chronic kidney disease, expressed as higher serum creatinine levels, was found to be a significant predictor for post-TAVI PM in the present study. This is in accordance to Sager and colleagues, who reported that the presence of chronic kidney disease was a risk factor for the occurrence of persistent conduction disturbances and for the subsequent requirement for PM [21]. Higher pre-procedural serum creatinine levels might also be associated with a persistent inflammatory status [22], which can negatively interact with the lesions induced by the valve implantation at the annular level, leading to an atrioventricular conduction defect. Furthermore, patients suffering from chronic kidney disease are usually more prone to extensive valvular calcification [23], which can play a role in the genesis of a heart conduction system injury following an aortic valve implant. However, no information was available on the chronicity of the kidney disease in our cohort.

A history of previous aortic valve surgery has been identified as a strong predictive factor for post-operative PM in patients undergoing surgical aortic valve replacement [24]. In this study, we found that previous aortic valve surgery was a protective factor for post-TAVI PM. Indeed, even if it seems logical to observe a higher PM rate after TAVI, due to the potential negative interaction by the TAVI procedure on pre-existent lesions caused by the previous surgical approach, the literature evidence remains scarce or controversial in this respect, as the valve-in-valve procedure has been associated with less annular stress on the conduction system, indicating that the previous implanted valve might protect against wall stress [25].

In the present study, more than three-quarters of the TAVI procedures were performed via a transfemoral approach, followed by transapical, transaortic, and trans-subclavian approaches. The transfemoral access was shown to represent an independent determinant of peri-procedural PM, with the lowest PM rate observed in patients undergoing transaortic access. This finding represents a novel insight in the understanding of the underlying predictors for post-TAVI PM if compared to previous investigations. Indeed, a few other studies have addressed such an association, yet failing to demonstrate any interplay between the antegrade versus retrograde approaches and post-TAVI PM. However, these studies were based on single-center experiences or meta-analysis [26,27]. Ewe and colleagues have described the transapical approach as a predictor of conduction disturbances after TAVI [28], but this relationship has not been confirmed by other studies [21,26].

Balloon post-dilatation was used in 20.1% of the patients in our cohort. The safety of balloon post-dilatation to improve clinical outcomes has been previously assessed [28,29]. However, balloon post-dilatation may increase the risk for post-operative PM, as shown in the analysis of Barbanti and colleagues [30]. A greater radial mechanical stress can be generated by the balloon dilatation procedure, further damaging the conduction system [28,29]; thereby, representing a possible causative mechanism to be considered. This is further highlighted by the higher rate of left bundle branch block after balloon post-dilatation [28], emphasizing the potential compression effect on the conduction branches and the atrioventricular node.

As emphasized by Kawsara and colleagues [30], in their nationwide analysis of the National Readmission Database in the United States, temporal and general trends of PM after TAVI remain poorly reported. We found a small and not significant reduction in post-TAVI PM across the study period. The trend in the United States has shown an increase in post-TAVI PM rates between 2012 and 2015, and then a decrease to a stable rate (around 10%) thereafter, in accordance with our findings [30]. The identification of potential patient- and technique-related predictors for post-TAVI PM may help to reduce the rate of post-TAVI PM across the time.

### Study Limitations

Our analysis presents several limitations. The main limitation is inherent to the retrospective and observational nature of the study, with the related biases characteristic of this methodology and to the collection of data in a registry. Additionally, due to missing PM status, 25% of the study population had to be excluded from the final studied population. We had no information about pacemaker indication, type provided, pre-existent conduction abnormalities or background therapy. Additionally, we could not obtain information about the TAVI device type and size, making us unable to draw any conclusions about the influence of these factors on PM, in particular in male subjects, or from the interpretation of the trend of post-TAVI permanent PM implantation over time. Nevertheless, this large national population-based analysis has provided a robust and real-life series of patients undergoing isolated TAVI procedure.

## 5. Conclusions

In conclusion, in the present study we analyzed the post-procedural rate of permanent PM implantation after TAVI in an entire country, demonstrating a stable incidence of this complication over a 6-year period. Creatinine serum level, age, weight, transfemoral TAVI approach, and TAVI pre-dilatation were found to be independent predictive factors for peri-procedural PM. Male sex and previous aortic valve surgery were found to be protective with respect to post-TAVI permanent PM implantation. In future studies, predictors such as these may be combined into a model for individual prediction of risk of permanent PM implantation and may be added to the informed consent process.

## Figures and Tables

**Figure 1 jcm-11-00560-f001:**
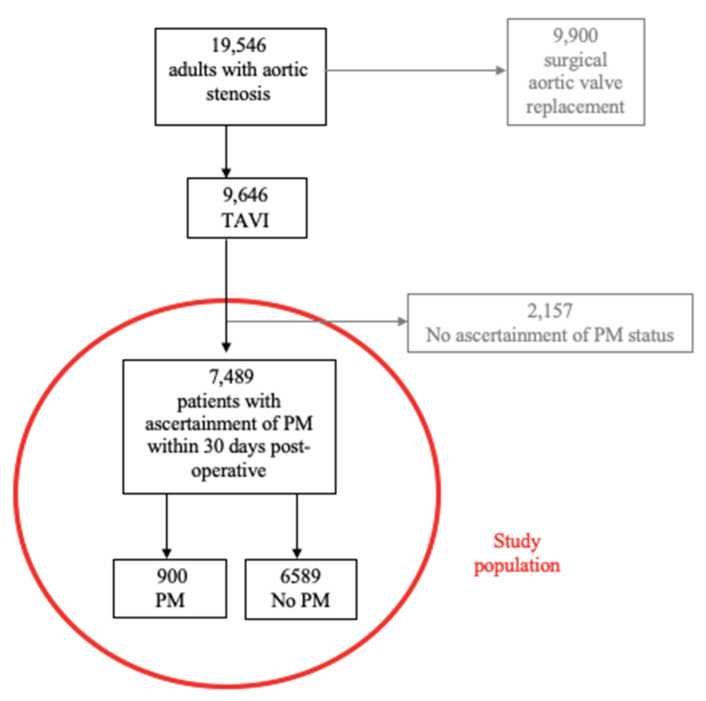
Flowchart of patients after transcatheter aortic valve implantation (TAVI) leading to permanent pacemaker (PM) implantation.

**Figure 2 jcm-11-00560-f002:**
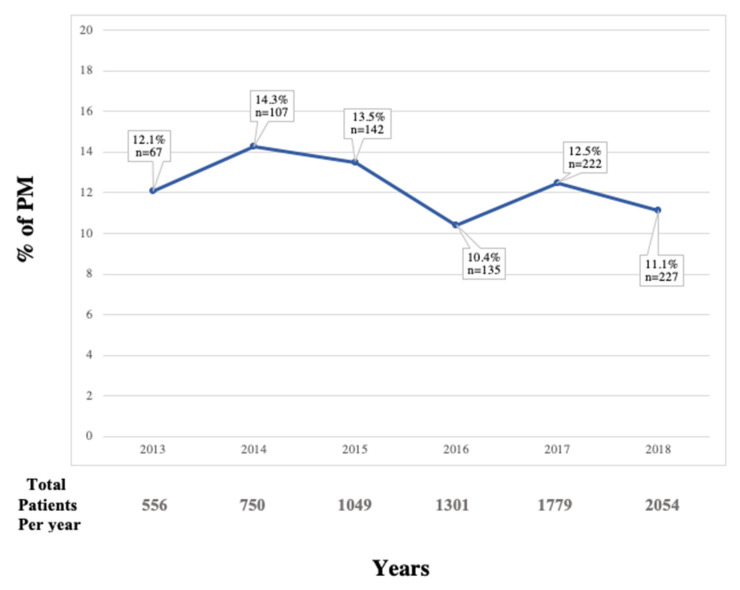
Trends of permanent pacemaker (PM) implementation after transcatheter aortic valve implantation (TAVI) over time.

**Table 1 jcm-11-00560-t001:** Baseline characteristics of patients with or without a 30-day permanent pacemaker (PM) implantation after transcatheter aortic valve implantation (TAVI).

Variables *	Total Study Group (n = 7489)	PM Group (n = 900)	No PM Group (n = 6589)
Male sex	3681 (49.2%)	494 (54.9%)	3187 (48.4%)
Age, mean (SD), y	79.7 (±6.8)	80.0 (±6.5)	79.7 (±6.9)
Weight, mean (SD), kg	77.2 (±15.8)	78.6 (±15.6)	77.1 (±15.7)
Creatinine serum level, median (IQR), μmol/L	92.0 (75.0–115.0)	95.0 (78.0–122.0)	92.0 (75.0–115.0)
Diabetes mellitus ^†^	2049 (27.6%)	258 (28.9%)	1791 (27.5%)
Left ventricular ejection fraction, median (IQR), %	55.0 (43.0–55.0)	55.0 (54.0–55.0)	55.0 (43.0–55.0)
Systolic pulmonary pressure, median (IQR), mmHg ^†^	25.0 (25.0–33.0)	25.0 (25.0–34.0)	25.0 (25.0–33.0)
History of lung disease ^†^	1624 (21.7%)	195 (21.7%)	1429 (21.7%)
Peripheral vascular disease ^†^	1654 (22.1%)	193 (21.5%)	1461 (22.2%)
Previous cardiac surgery	1505 (20.7%)	160 (18.5%)	1345 (21%)
Recent myocardial infarction	141 (1.9%)	14 (1.6%)	127 (1.9%)
Dialysis	96 (1.3%)	16 (1.8%)	80 (1.2%)
Euroscore II, median (IQR), %	3.6 (2.2–6.1)	3.7 (2.2–6.1)	3.6 (2.2–6.1)
Previous aortic valve surgery	339 (4.6%)	19 (2.1%)	320 (4.9%)

* Numbers are presented as valid percentages, excluding missing values. ^†^ See Appendix A for definitions of baseline characteristics Values are n (%). SD = standard deviation; IQR = interquartile range; PM = pacemaker.

**Table 2 jcm-11-00560-t002:** Procedural characteristics of patients with or without a 30-day permanent pacemaker (PM) implementation after transcatheter aortic valve implantation (TAVI).

Variables *	Total Study Group	PM Group	No PM Group
TAVI access	(n = 7489)	(n = 900)	(n = 6589)
Transfemoral surgical	1036 (14.0%)	98 (11.9%)	938 (14.4%)
Transfemoral percutaneous	4218 (57.2%)	548 (60.9%)	3670 (56.4%)
Transfemoral unknown	552 (7.5%)	90 (10.3%)	462 (7.1%)
Trans-subclavian	420 (5.7%)	44 (4.9%)	376 (5.8%)
Transapical	568 (7.7%)	49 (5.4%)	519 (8.0%)
Direct transaortic	562 (7.6%)	43 (4.8%)	519 (8.0%)
Pre-dilatation valve	3544 (49.5%)	402 (47.3%)	3142 (49.7%)
Post-dilatation valve	1029 (14.9%)	165 (20.1%)	864 (14.2%)

* Numbers are presented as valid percentage, excluding missing values. Values are n (%). PM = pacemaker; TAVI = transcatheter aortic valve implantation.

**Table 3 jcm-11-00560-t003:** Multivariate analysis of a 30-day permanent pacemaker (PM) implantation after transcatheter aortic valve implantation (TAVI).

Variables	OR (95% CI)	*p* Value
Male sex	0.80 (0.68–0.93)	<0.01
Age (years)	1.01 (1.00–1.02)	0.08
Weight (kg)	1.00 (1.00–1.01)	0.10
Creatinine serum level (for 100 μmol/L)	1.15 (1.01–1.31)	0.04
Diabetes mellitus	1.04 (0.88–1.22)	0.65
Left ventricular ejection fraction (%)	1.00 (1.00–1.01)	0.43
Systolic pulmonary pressure (mmHg)	1.00 (1.00–1.01)	0.51
History of lung disease	1.00 (0.84–1.19)	0.97
Peripheral vascular disease	0.98 (0.81–1.19)	0.87
Previous cardiac surgery	0.95 (0.74–1.20)	0.64
Recent myocardial infarction	0.80 (0.46–1.41)	0.44
Dialysis	0.84 (0.38–1.88)	0.67
Euroscore II (%)	1.00 (0.98–1.02)	0.96
Previous aortic valve surgery	0.42 (0.26–0.69)	<0.01
Transfemoral TAVI access	1.34 (1.11–1.61)	<0.01
Pre-dilatation valve	0.90 (0.78–1.03)	0.14
Post-dilatation valve	1.58 (1.33–1.89)	<0.01

OR = odds ratio; CI = confidence interval; TAVI = transcatheter aortic valve implantation.

## Data Availability

The data underlying this article will be shared on reasonable request to the corresponding author.

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
