# Peer review of "Incidence and Predictors of Permanent Pacemaker Implantation after Transcatheter Aortic Valve Procedures: Data of The Netherlands Heart Registration (NHR)"

_jcm, 2022, doi:10.3390/jcm11030560_

Round 1

Reviewer 1 Report

In this manuscript, Ravaux JM et al. present data on a long-term, retrospective analysis of patients undergoing TAVR in the Netherlands. They demonstrate that implantation of a permanent pacemaker is a frequent complication after TAVR, and present statistical modeling implicating several patient- and procedure-related risk factors independently predicting PM implantation.

The manuscript is generally written, with only minor spelling and phrasing issues. I find the discussion well composed, explaining the potential reasons for the role of the discovered risk factors.

Major points

The authors might consider giving exact numbers on the frequency of PM after SAVR right in the beginning of the introduction, as this would make the contrast (or lack thereof) more easily apparent to the reader.

Do the authors have information on the temporal relationship between TAVR and previous coronary revascularization. Likewise, do they have information on the frequency on CAD in general (not MI only), and which vessel was revascularized before TAVR?

The authors should add the information that 19546 patients were hospitalized for severe AS to Figure 1, delineating how patients were treated that were not included in the population of 9646 isolated TAVR patients.

Table 1: is background therapy at the time of hospitalization for isolated TAVI available? Might this influence outcomes, e.g. intake of beta blockers and the risk for requiring permanent PM? How were diabetes mellitus, history of lung disease, PAD, etc. defined? Is there information on the frequency of arterial hypertension and dyslipidemia? How was sysPAP measured, by echocardiography?

For renal failure, I would opt against the use of serum creatinine, as I only incompletely reflects changes in actual renal function in high-GFR ranges. Likewise, might serum creatinine reflect acute kidney injury rather than chronic renal failure in some patients? Longitudinal comparison whether pre-TAVR serum creatinine levels are due to acute or chronic disease should be considered.  

Minor points

27: “We aimed…”

36-39: The information given in this section of the abstract is the same as in the few lines before, but without numbers. Please consider finding a different conclusion here.

66: “In this study, we aimed…”

70: “which registers…”

110: In Figure 1, the abbreviation “PPI” is used, making this the only section of the paper.

Table 2: left-align the first column of the table.

148: “factor”, “predictor”

207: “Furthermore, patients…”

229: “… confirmed by other studies.”

245: “Identification of potential patient- and technique-related predictors…”

250: “…characteristic for…”

254: “…we could not…”

262: “… of this complication over a 6-year period.”

262: “… age, weight, trans-femoral…”

Author Response

In this manuscript, Ravaux JM et al. present data on a long-term, retrospective analysis of patients undergoing TAVR in the Netherlands. They demonstrate that implantation of a permanent pacemaker is a frequent complication after TAVR, and present statistical modeling implicating several patient- and procedure-related risk factors independently predicting PM implantation.

The manuscript is generally written, with only minor spelling and phrasing issues. I find the discussion well composed, explaining the potential reasons for the role of the discovered risk factors.

Major points

  1. The authors might consider giving exact numbers on the frequency of PM after SAVR right in the beginning of the introduction, as this would make the contrast (or lack thereof) more easily apparent to the reader.

Reply: Thank you for this pertinent comment. The average post-operative rate of pacemaker implantation after surgical aortic valve replacement (2 to 6%) has been added to the introduction of the manuscript, to highlight the contrast between the two techniques.

Changes: See manuscript, page 2, line 56.

  1. Do the authors have information on the temporal relationship between TAVR and previous coronary revascularization. Likewise, do they have information on the frequency on CAD in general (not MI only), and which vessel was revascularized before TAVR?

Reply: Thank you for this interesting comment. The NHR registry is divided into a surgical part and a cardiological part for the record of the data. Unfortunately, data about previous coronary revascularization, coronary artery disease and eventual revascularization before TAVR were not available in the surgical part of the national database. A further manuscript, written in collaboration with the cardiologist of Maastricht will analyze this interesting issue.

Changes: None.

  1. The authors should add the information that 19546 patients were hospitalized for severe AS to Figure 1, delineating how patients were treated that were not included in the population of 9646 isolated TAVR patients.

Reply: Thank you for this important comment. The figure 1 has been modified, in order to better represent the flowsheet of the selection of the study population.

Changes:  See manuscript, page 3, line 108.

  1. Table 1: is background therapy at the time of hospitalization for isolated TAVI available? Might this influence outcomes, e.g. intake of beta blockers and the risk for requiring permanent PM? How were diabetes mellitus, history of lung disease, PAD, etc. defined? Is there information on the frequency of arterial hypertension and dyslipidemia? How was sysPAP measured, by echocardiography?

Reply: Thank you for this pertinent comment. Unfortunately, we had no data about background therapy at the time of the hospitalization. As the pre-operative rhythmic therapy may also help to identify potential high-risk patients for post-operative conduction disorders, it would have been interesting to describe the baseline pharmacological therapy of this cohort. This was added to the limitations section of the manuscript. Also, the definition of the diabetes mellitus, history of lung disease, PAD and measure of systolic pulmonary pressure were added in the supplemental materials, addendum 2. Information about arterial hypertension and dyslipidemia were unfortunately not available in the register.

Changes: See manuscript, page 7, lines 254-255 and supplemental material, addendum 2.

  1. For renal failure, I would opt against the use of serum creatinine, as I only incompletely reflects changes in actual renal function in high-GFR ranges. Likewise, might serum creatinine reflect acute kidney injury rather than chronic renal failure in some patients? Longitudinal comparison whether pre-TAVR serum creatinine levels are due to acute or chronic disease should be considered.  

Reply: Thank you for this interesting comment. Indeed, as the metabolism of creatinine is affected, among others, by the creatine level in the muscles and the secretion of creatinine in the tubules, change in serum creatinine rates may not always reflect the deterioration in kidney function, without indication on the chronicity of these condition. However, the median age of this cohort was 79,7 years and eGFR calculation may also be less reliable in older patients (Raman et al. Estimating renal function in old people: an in-depth review. Int Urol Nephrol 2017; 49(11):1979-1988). Therefore, we emphasized that higher pre-procedural creatinine level may reasonably be correlated to a persistent inflammatory status, explaining partially the potential higher risk for post-operative rhythm disturbances. However, a sentence was added in the discussion to emphasize that no information was available on the chronicity of the kidney disease in this cohort.

Changes: See manuscript, page 7, lines 216-217.

Minor points

  1. 27: “We aimed…”

Reply: Thank you for this pertinent comment. The sentence has been corrected.

Changes: See manuscript, page 1, line 27.

  1. 36-39: The information given in this section of the abstract is the same as in the few lines before, but without numbers. Please consider finding a different conclusion here.

Reply: Thank you for this pertinent comment. A more appropriate sentence has been added to the conclusion of the abstract, underlying the importance of reducing post-TAVI PM rate and identifying high-risk patients.

Changes: See manuscript, page 1, 36-39

  1. 66: “In this study, we aimed…”

Reply: Thank you for this pertinent comment. The sentence has been corrected.

Changes: See manuscript, page 2, line 67.

  1. 70: “which registers…”

Reply: Thank you for this pertinent comment. The sentence has been corrected.

Changes: See manuscript, page 2, line 71.

  1. 110: In Figure 1, the abbreviation “PPI” is used, making this the only section of the paper.

Reply: Thank you for this important comment. The figure 1 has been modified in order to use the same abbreviation throughout the manuscript.

Changes: See manuscript, page 3, line 108, Figure 1.  

  1. Table 2: left-align the first column of the table.

Reply: Thank you for this judicious comment. The first-column of the table has been modified.

Changes: See manuscript, page 4, line 128, Table 2.

  1. 148: “factor”, “predictor”

Reply: Thank you for this pertinent comment. The sentence has been corrected.

Changes: See manuscript, page 5, line 155.

  1. 207: “Furthermore, patients…”

Reply: Thank you for this pertinent comment. The sentence has been corrected.

Changes: See manuscript, page 6, line 214.

  1. 229: “… confirmed by other studies.”

Reply: Thank you for this pertinent comment. The sentence has been corrected.

Changes: See manuscript, page 7, line 237.

  1. 245: “Identification of potential patient- and technique-related predictors…”

Reply: Thank you for this pertinent comment. The sentence has been corrected.

Changes: See manuscript, page 7, lines 253-255.

  1. 250: “…characteristic for…”

Reply: Thank you for this pertinent comment. The sentence has been corrected.

Changes: See manuscript, page 7, line 258.

  1. 254: “…we could not…”

Reply: Thank you for this pertinent comment. The sentence has been corrected.

Changes: See manuscript, page 7, line 262.

  1. 262: “… of this complication over a 6-year period.”

Reply: Thank you for this pertinent comment. The sentence has been corrected.

Changes: See manuscript, page 7, lines 269-271.

  1. 262: “… age, weight, trans-femoral…”

Reply: Thank you for this pertinent comment. The sentence has been corrected.

Changes: See manuscript, page 7, line 272.

Reviewer 2 Report

Dr. Ravaux and co-workers show in their large retrospective data analysis of the Netherland Heart Registration in 7489 factors which are related to the implantation of a pacemaker following TAVI procedure. The predictors of the need of pacemaker implantation following TAVI was age, weight, impaired renal function, femoral approach and TAVI post dilatation. These are in line with most of the published factors responsible for the need of pacemaker implantation thus far. Especially the large number and long follow-up speak in favor of the reported data.

From the perspective of the reviewer there are no specific points the authors should address in the current manuscript.

Author Response

Dr. Ravaux and co-workers show in their large retrospective data analysis of the Netherland Heart Registration in 7489 factors which are related to the implantation of a pacemaker following TAVI procedure. The predictors of the need of pacemaker implantation following TAVI was age, weight, impaired renal function, femoral approach and TAVI post dilatation. These are in line with most of the published factors responsible for the need of pacemaker implantation thus far. Especially the large number and long follow-up speak in favor of the reported data.

  1. From the perspective of the reviewer there are no specific points the authors should address in the current manuscript

Reply: Thank you for this positive comment.

Changes: None